# Robustness of Bayesian Neural Networks to Gradient-Based Attacks

**Ginevra Carbone***
Department of Mathematics and Geosciences
University of Trieste, Trieste, Italy
ginevra.carbone@phd.units.it

**Matthew Wicker***
Departement of Computer Science
University of Oxford,
Oxford, United Kingdom
matthew.wicker@wolfson.ox.ac.uk

**Luca Laurenti**
Departement of Computer Science,
University of Oxford,
Oxford, United Kingdom
luca.laurenti@cs.ox.ac.uk

**Andrea Patane**
Departement of Computer Science,
University of Oxford,
Oxford, United Kingdom
patane.andre@gmail.com

**Luca Bortolussi**
Department of Mathematics and Geosciences
University of Trieste, Trieste, Italy;
Modeling and Simulation Group,
Saarland University, Saarland, Germany
luca.bortolussi@gmail.com

**Guido Sanguinetti**
School of Informatics, University
of Edinburgh, Edinburgh, United Kingdom;
SISSA, Trieste, Italy
gsanguin@inf.ed.ac.uk

## Abstract

Vulnerability to adversarial attacks is one of the principal hurdles to the adoption of deep learning in safety-critical applications. Despite significant efforts, both practical and theoretical, the problem remains open. In this paper, we analyse the geometry of adversarial attacks in the large-data, overparametrized limit for Bayesian Neural Networks (BNNs). We show that, in the limit, vulnerability to gradient-based attacks arises as a result of degeneracy in the data distribution, i.e., when the data lies on a lower-dimensional submanifold of the ambient space. As a direct consequence, we demonstrate that in the limit BNN posteriors are robust to gradient-based adversarial attacks. Experimental results on the MNIST and Fashion MNIST datasets, representing the finite data regime, with BNNs trained with Hamiltonian Monte Carlo and Variational Inference support this line of argument, showing that BNNs can display both high accuracy and robustness to gradient based adversarial attacks.

## 1 Introduction

Adversarial attacks are small, potentially imperceptible pertubations of test inputs that can lead to catastrophic misclassifications in high-dimensional classifiers such as deep Neural Networks (NN). Since the seminal work of Szegedy et al. [2013], adversarial attacks have been intensively studied, and even state-of-the-art deep learning models, trained on very large data sets, have been shown to be susceptible to such attacks [Goodfellow et al., 2014]. In the absence of effective defenses, the widespread existence of adversarial examples has raised serious concerns about the security and robustness of models learned from data [Biggio and Roli, 2018]. As a consequence, the development of machine learnig models that are robust to adversarial perturbations is an essential pre-condition for

their application in safety-critical scenarios, where model failures have already led to fatal accidents [Yadron and Tynan, 2016].

Many attack strategies are based on identifying directions of high variability in the loss function by evaluating gradients w.r.t. input points (see, e.g., Goodfellow et al. [2014], Madry et al. [2017]). Since such variability can be intuitively linked to uncertainty in the prediction, Bayesian Neural Networks (BNNs) [Neal, 2012] have been recently suggested as a more robust deep learning paradigm, a claim that has found some empirical support [Feinman et al., 2017, Gal and Smith, 2018, Bekasov and Murray, 2018, Liu et al., 2018]. However, neither the source of this robustness, nor its general applicability are well understood mathematically.

In this paper we show a remarkable property of BNNs: in a suitably defined large data limit, we prove that the gradients of the expected loss function of a BNN w.r.t. the input points vanish. Our analysis shows that adversarial attacks for deterministic NNs in the large data limit arise necessarily from the low dimensional support of the data generating distribution. By averaging over nuisance dimensions, BNNs achieve zero expected gradient of the loss and are thus provably immune to gradient-based adversarial attacks.

We experimentally support our theoretical findings on various BNN architectures trained with Hamiltonian Monte Carlo (HMC) and with Variational Inference (VI) on both MNIST and Fashion MNIST data sets, empirically showing that the magnitude of the gradients decreases as more samples are taken from the BNN posterior. We also test this decreasing effect when approaching towards the overparametrized case on the Half Moons dataset. We experimentally show that two popular gradient-based attack strategies for attacking NNs are unsuccessful on BNNs. Finally, we conduct a large-scale experiment on thousands of different networks, showing that for BNNs high accuracy correlates with high robustness to gradient-based adversarial attacks, contrary to what observed for deterministic NNs trained via standard Stochastic Gradient Descent (SGD) [Su et al., 2018].

In summary, this paper makes the following contributions:

- A theoretical framework to analyse adversarial robustness of BNNs in the large data limit.
- A proof that, in this limit, the posterior average of the gradients of the loss function vanishes, providing robustness against gradient-based attacks.
- Large-scale experiments, showing empirically that BNNs are robust to gradient-based attacks and can resist the well known accuracy-robustness trade-off.[1]

**Related Work**   The robustess of BNNs to adversarial examples has been already observed by Gal and Smith [2018], Bekasov and Murray [2018]. In particular, in [Bekasov and Murray, 2018] the authors define Bayesian adversarial spheres and empirically show that, for BNNs trained with HMC, adversarial examples tend to have high uncertainty, while in [Gal and Smith, 2018] sufficient conditions for idealised BNNs to avoid adversarial examples are derived. However, it is unclear how such conditions could be checked in practice, as it would require one to check that the BNN architecture is invariant under all the symmetries of the data.

Empirical methods to detect adversarial examples for BNNs that utilise pointwise uncertainty have been introduced in [Li and Gal, 2017, Feinman et al., 2017, Rawat et al., 2017]. However, most of these approaches have largely relied on Monte Carlo dropout for posterior inference [Carlini and Wagner, 2017]. Statistical techniques for the quantification of adversarial robustness of BNNs have been introduced by [Cardelli et al., 2019a] and employed in [Michelmore et al., 2019] to detect erroneous behaviours in the context of autonomous driving. Furthermore, in [Ye and Zhu, 2018] a Bayesian approach has been considered in the context of adversarial training, where the authors showed improved performances with respect to other, non-Bayesian, adversarial training approaches.

## 2   Bayesian Neural Networks and Adversarial Attacks

Bayesian modelling aims to capture the intrinsic epistemic uncertainty of data driven models by defining ensembles of predictors [Barber, 2012]; it does so by turning algorithm parameters (and consequently predictions) into random variables. In the NN scenario, for a NN $f(\mathbf{x}, \mathbf{w})$ with input $\mathbf{x}$ and network parameters (weights) $\mathbf{w}$, one starts with a prior measure over the network weights $p(\mathbf{w})$

[Neal, 2012]. The fit of the network with weights $\mathbf{w}$ to the data $D$ is assessed through the likelihood $p(D|\mathbf{w})$ [Bishop, 2006]. Bayesian inference then combines likelihood and prior via Bayes theorem to obtain a *posterior* measure on the space of weights $p\left(\mathbf{w}|D\right) \propto p\left(D|\mathbf{w}\right)p\left(\mathbf{w}\right)$.

Maximising the likelihood function w.r.t. the weights $\mathbf{w}$ is in general equivalent to minimising the loss function in standard NNs; indeed, standard training of NNs can be viewed as an approximation to Bayesian inference which replaces the posterior distribution with a delta function at its mode. Obtaining the posterior distribution exactly is impossible for non-linear/non-conjugate models such as NNs. Asymptotically exact samples from the posterior distribution can be obtained via procedures such as Hamiltonian Monte Carlo (HMC) [Neal et al., 2011], while approximate samples can be obtained more cheaply via Variational Inference (VI) [Blundell et al., 2015]. Irrespective of the posterior inference method of choice, Bayesian predictions at a new input $\mathbf{x}^*$ are obtained from an ensemble of $n$ NNs, each with its individual weights drawn from the posterior distribution $p(\mathbf{w}|D)$ :

$$f\left(\mathbf{x}^*|D\right) = \langle f(\mathbf{x}^*,\mathbf{w})\rangle_{p(\mathbf{w}|D)} \simeq \frac{1}{n}\sum_{i=1}^{n} f(\mathbf{x}^*,\mathbf{w}_i) \qquad \mathbf{w}_i \sim p\left(\mathbf{w}|D\right) \qquad (1)$$

where $\langle\cdot\rangle_p$ denotes expectation w.r.t. the distribution $p$. The ensemble of NNs yields the *predictive distribution* of the BNN.

Given an input point $\mathbf{x}^*$ and a strength (i.e. maximum perturbation magnitude) $\epsilon > 0$, the worst-case adversarial perturbation can be defined as the point around $\mathbf{x}^*$ that maximises the loss function $L(\tilde{\mathbf{x}},\mathbf{w})$[2]:

$$\tilde{\mathbf{x}} := \underset{\tilde{\mathbf{x}}:||\tilde{\mathbf{x}}-\mathbf{x}^*||\leq\epsilon}{\arg\max}\ \langle L(\tilde{\mathbf{x}},\mathbf{w})\rangle_{p(\mathbf{w}|D)}.$$

If the network prediction on $\tilde{\mathbf{x}}$ differs from the original prediction on $\mathbf{x}^*$, then we call $\tilde{\mathbf{x}}$ an *adversarial example*. As $f(\mathbf{x},\mathbf{w})$ is non-linear, computing $\tilde{\mathbf{x}}$ is a non-linear optimisation problem for which several approximate solution methods have been proposed and among them, gradient-based attacks are arguably the most prominent [Biggio and Roli, 2018]. In particular, the Fast Gradient Sign Method (FGSM) [Goodfellow et al., 2014] is among the most commonly employed attacks and works by approximating $\tilde{\mathbf{x}}$ by taking an $\epsilon$-step in the direction of the sign of the gradient at $\mathbf{x}$. In the context of BNNs, where attacks are against the predictive distribution of Eqn (1), FGSM becomes

$$\tilde{\mathbf{x}} \simeq \mathbf{x} + \epsilon\,\mathrm{sgn}\left(\langle\nabla_{\mathbf{x}}L(\mathbf{x},\mathbf{w})\rangle_{p(\mathbf{w}|D)}\right) \simeq \mathbf{x} + \epsilon\,\mathrm{sgn}\left(\sum_{i=1}^{n}\nabla_{\mathbf{x}}L(\mathbf{x},\mathbf{w}_i)\right) \qquad (2)$$

where the final expression is a Monte Carlo approximation with samples $\mathbf{w}_i$ drawn from the posterior $p(\mathbf{w}|D)$. The expressions for the Projected Gradient Descent method (PGD) [Madry et al., 2017] or other gradient-based attacks are analogous. While the results discussed in Section 3 hold for any gradient-based method, in the experiments reported in Section 5 we focus on the Fast Gradient Sign Method (FGSM) [Goodfellow et al., 2014] and the Projected Gradient Descent method (PGD) [Madry et al., 2017].

## 3 Adversarial robustness of Bayesian predictive distributions

Equation (2) suggests a possible explanation for the observed robustness of BNNs to adversarial attacks: the averaging under the posterior might lead to cancellations in the final expectation of the gradient. It turns out that this averaging property is intimately related to the geometry of the so called *data manifold* $\mathcal{M}_D \subset \mathbb{R}^d$, i.e. the support of the data generating distribution $p(D)$. The key result that we leverage is a recent breakthrough [Du et al., 2018, Rotskoff and Vanden-Eijnden, 2018, Mei et al., 2018] which proved global convergence of (stochastic) gradient descent (at the distributional level) in the overparametrised, large data limit. Precise definitions can be found in the original publications and in the supplementary material. In our setting, a *fully trained, overparametrized BNN* is an ensemble of NNs satisfying the conditions in [Rotskoff and Vanden-Eijnden, 2018] and at full convergence of the training algorithm, hence they all coincide on the data manifold, but can differ outside of it. We now state our main result whose full proof is in the supplementary material:

**Theorem 1.** *Let $f(\mathbf{x}, \mathbf{w})$ be a fully trained overparametrized BNN on a prediction problem with data manifold $\mathcal{M}_D \subset \mathbb{R}^d$ and posterior weight distribution $p(\mathbf{w}|D)$. Assuming $\mathcal{M}_D \in \mathcal{C}^\infty$ almost everywhere, in the large data limit we have a.e. on $\mathcal{M}_D$*

$$\left( \langle \nabla_\mathbf{x} L(\mathbf{x}, \mathbf{w}) \rangle_{p(\mathbf{w}|D)} \right) = \mathbf{0}. \tag{3}$$

By the definition of the FGSM attack (Equation (2)) and other gradient-based attacks, Theorem 1 directly implies that any gradient-based attack will be ineffective against a BNN in the limit. The theorem is proved by first showing that in a fully trained BNN in the large data limit, the gradient of the loss is orthogonal to the data manifold (Lemma 1 and Corollary 1), then proving a symmetry property of a fully trained BNN with an uninformative prior, guaranteeing that the orthogonal component of the gradient cancels out in expectation with respect to the BNN posterior (Lemma 2).

**Dimensionality of the data manifold**    To investigate the effect of dimensionality of the data manifold on adversarial examples, we start from the following

**Lemma 1.** *Let $f(\mathbf{x}, \mathbf{w})$ be a fully trained overparametrized NN on a prediction problem with a.e. smooth data manifold $\mathcal{M}_D \subset \mathbb{R}^d$. Let $\mathbf{x}^* \in \mathcal{M}_D$ s.t. $B_d(\mathbf{x}^*, \epsilon) \subset \mathcal{M}_D$, with $B_d(\mathbf{x}^*, \epsilon)$ being the $d$-dimensional ball centred at $\mathbf{x}^*$ of radius $\epsilon$ for some $\epsilon > 0$. Then $f(\mathbf{x}, \mathbf{w})$ is robust to gradient-based attacks at $\mathbf{x}^*$ of strength $\leq \epsilon$ (i.e. restricted in $B_d(\mathbf{x}^*, \epsilon)$).*

This is a trivial consequence of an important result proved in [Du et al., 2018, Rotskoff and Vanden-Eijnden, 2018, Mei et al., 2018]: at convergence, overparametrised NNs provably achieve zero loss on the whole data manifold $\mathcal{M}_D$ in the infinite data limit, which implies that the function $f$ would be locally constant at $\mathbf{x}^*$. A corollary of Lemma 1 is

**Corollary 1.** *Let $f(\mathbf{x}, \mathbf{w})$ be a fully trained overparametrized NN on a prediction problem with data manifold $\mathcal{M}_D \subset \mathbb{R}^d$ smooth a.e. (where the measure is given by the data distribution $p(D)$). If $f$ is vulnerable to gradient-based attacks at $x^* \in \mathcal{M}_D$ in the infinite data limit, then a.s. $\dim(\mathcal{M}_D) < d$ in a neighbourhood of $x^*$.*

This corollary confirms the widely held conjecture that adversarial attacks may originate from degeneracies of the data manifold [Goodfellow et al., 2014, Fawzi et al., 2018]. In fact, it had been already empirically noticed [Khoury and Hadfield-Menell, 2018] that adversarial perturbations often arise in directions which are normal to the data manifold. The higher the codimension of the data manifold into the embedding space, the more it is likely to select random directions which are normal to it. The suggestion that lower-dimensional data structures might be ubiquitous in NN problems is also corroborated by recent results [Goldt et al., 2019] showing that the characteristic training dynamics of NNs are intimately linked to data lying on a lower-dimensional manifold. Notice that the implication is only one way; it is perfectly possible for the data manifold to be low dimensional and still not vulnerable at many points.

Notice that the assumption of smoothness a.e. for the data manifold is needed to avoid pathologies in the data distribution (e.g. its support being a closed but dense subset of $\mathbb{R}^d$). Additionally, this assumption guarantees that the dimensionality of $\mathcal{M}_D$ is locally constant. A consequence of Corollary 1 is that $\forall \mathbf{x} \in \mathcal{M}_D$ the gradient of the loss function is orthogonal to the data manifold as it is zero along the data manifold, i.e., $\nabla_\mathbf{x} L(\mathbf{x}, \mathbf{w}) = \nabla_{\perp\mathbf{x}} L(\mathbf{x}, \mathbf{w})$, where $\nabla_{\perp\mathbf{x}}$ denotes the gradient projected into the normal subspace of $\mathcal{M}_D$ at $\mathbf{x}$.

**Bayesian averaging of normal gradients**    In order to complete the proof of Theorem 1, we therefore need to show that the normal gradient has expectation zero under the posterior distribution

$$\nabla_{\perp\mathbf{x}} \langle L(\mathbf{x}, \mathbf{w}) \rangle_{p(\mathbf{w}|D)} = 0.$$

The key to this result is the fact that, assuming an uninformative prior[3] on the weights $\mathbf{w}$, all NNs that agree on the data manifold will by definition receive the same posterior weight in the ensemble, since they achieve exactly the same likelihood. Therefore, it remains to be proved the following symmetry of the normal gradient at almost any point $\hat{\mathbf{x}} \in \mathcal{M}_D$:

**Lemma 2.** *Let $f(\mathbf{x}, \mathbf{w})$ be a fully trained overparametrized NN on a prediction problem on data manifold $\mathcal{M}_D \subset \mathbb{R}^d$ a.e. smooth. Let $\hat{\mathbf{x}} \in \mathcal{M}_D$ to be the perturbed input and let the normal gradient at $\hat{\mathbf{x}}$ be $\mathbf{v}_{\mathbf{w}}(\hat{\mathbf{x}}) = \nabla_{\perp \hat{\mathbf{x}}} L(\mathbf{x}, \mathbf{w})$ be different from zero. Then, in the infinite data limit and for almost all $\hat{\mathbf{x}}$, there exists a set of weights $\mathbf{w}'$ such that*

$$f(\mathbf{x}, \mathbf{w}') = f(\mathbf{x}, \mathbf{w}) \text{ a.e. in } \mathcal{M}_D, \tag{4}$$

$$\nabla_{\perp \hat{\mathbf{x}}} L(\hat{\mathbf{x}}, \mathbf{w}') = -\mathbf{v}_{\mathbf{w}}(\hat{\mathbf{x}}). \tag{5}$$

The proof of this lemma rests on constructing a function satisfying (4) and (5) by suitably perturbing locally the fully trained NN $f(\mathbf{x}, \mathbf{w})$, i.e. by adding a function $\phi$ which is zero on the data manifold and enforces condition (5) on $\hat{\mathbf{x}}$. Since we are in the overparametrized, large data limit, any such function will be realisable as a NN with suitable weights choice $\mathbf{w}'$.

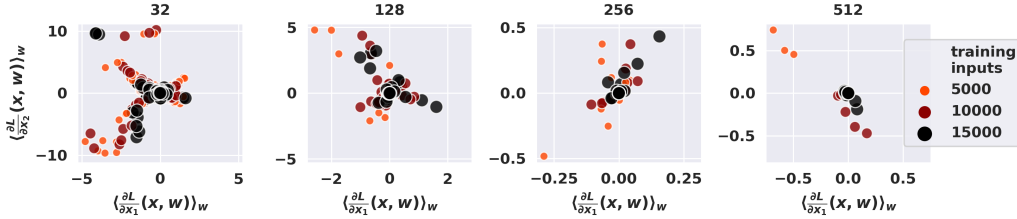

Figure 1: Expected loss gradients components on 100 two-dimensional test points from the Half Moons dataset [Rozza et al., 2014] (both partial derivatives of the loss function are shown). Each dot represents a different NN architecture. We used a collection of HMC BNNs, by varying the number of hidden neurons and training points. Only models with test accuracy greater than 80% were taken into account. We refer the reader to the supplementary material for training hyperparameters.

## 4 Consequences and limitations

Theorem 1 has the natural consequence of protecting BNNs against all gradient-based attacks, due to the vanishing average of the expectation of the gradients in the limit. Its proof also sheds light on a number of observations made in recent years. Before moving on to empirically validating the theorem, it is worth reflecting on some of its implications and limitations:

- Theorem 1 holds in a specific thermodynamic limit, however we expect the averaging effect of BNN gradients to still provide considerable protection in conditions where the network architecture and the data amount lead to high accuracy and strong expressivity. In practice, high accuracy might be a good indicator of robustness for BNNs. In Figure 1, we examine the impact of the assumptions made in Theorem 1 by exploiting a setting in which we have access to the data-generating distribution, the half-moons dataset [Rozza et al., 2014]. We show that the magnitude of the expectation of the gradient shrinks as we increase the network's parameters and the number of training inputs.
- Theorem 1 holds when the ensemble is drawn from the true posterior; nevertheless it is not obvious (and likely not true) that the posterior distribution is the sole ensemble with the zero averaging property of the gradients. Cheaper approximate Bayesian inference methods which retain ensemble predictions such as VI may in practice provide good protection.
- Theorem 1 is proven under the assumption of uniform priors; in practice, (vague) Gaussian priors are more frequently used for computational reasons. Once again, unless the priors are too informative, we do not expect a major deviation from the idealised case.
- Gaussian Processes [Williams and Rasmussen, 2006] are equivalent to infinitely wide BNNs and therefore constitute overparametrized BNNs by definition (although scaling their training to the large data limit might be problematic). Theorem 1 provides theoretical backing to recent empirical observations of their adversarial robustness [Blaas et al., 2019, Cardelli et al., 2019b].
- While the Bayesian posterior ensemble may not be the only randomization to provide protection, it is clear that some simpler randomizations such as bootstrap will be ineffective, as noted

empirically in [Bekasov and Murray, 2018]. This is because bootstrap resampling introduces variability along the data manifold, rather than in orthogonal directions. In this sense, the often repeated mantra that bootstrap is an approximation to Bayesian inference is strikingly inaccurate when the data distribution has zero measure support. Similarly, we do not expect gradient smoothing approaches to be successful [Athalye et al., 2018], since the type of smoothing performed by Bayesian inference is specifically informed by the geometry of the data manifold.

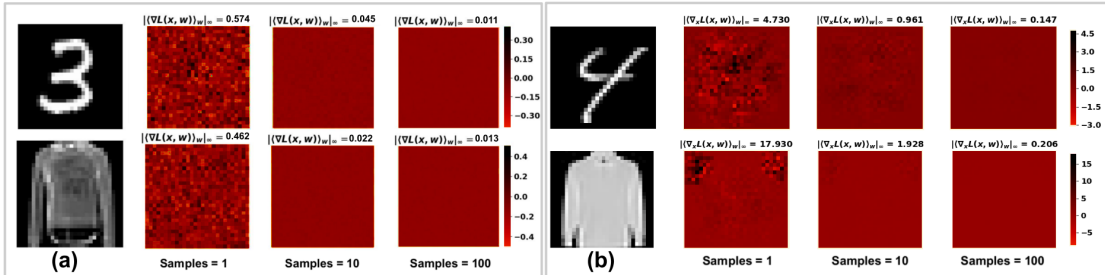

Figure 2: The expected loss gradients of BNNs exhibit a vanishing behaviour when increasing the number of samples from the posterior predictive distribution. We show example images from MNIST (top row) and Fashion MNIST (bottom row) and their expected loss gradients wrt networks trained with HMC (left) and VI (right). To the right of the images we plot a heat map of gradient values.

## 5 Empirical Results

In this section we empirically investigate our theoretical findings on different BNNs. We train a variety of BNNs on the MNIST and Fashion MNIST [Xiao et al., 2017] datasets, and evaluate their posterior distributions using HMC and VI approximate inference methods. In Section 5.1, we experimentally verify the validity of the zero-averaging property of gradients implied by Theorem 1, and discuss its implications on the behaviours of FGSM and PGD attacks on BNNs in Section 5.2. In Section 5.3 we analyse the relationship between robustness and accuracy on thousands of different NN architectures, comparing the results obtained by Bayesian and by deterministic training. Details on the experimental settings and BNN training parameters can be found in the Supplementary Material.

### 5.1 Evaluation of the Gradient of the Loss for BNNs

We investigate the vanishing behavior of input gradients - established by Theorem 1 for the thermo-dynamic limit regime - in the finite, practical settings, that is with a finite number of training data and with finite-width BNNs. Specifically, we train a two hidden layers BNN (with 1024 neurons per layer for a total of about $1.8$ million parameters) with HMC and a three hidden layers BNN (512 neurons per layer) with VI. These achieve approximately $95\%$ test accuracy on MNIST and $89\%$ on Fashion MNIST when trained with HMC; as well as $95\%$ and $92\%$, respectively, when trained with VI. More details about the hyperparameters used for training can be found in the Supplementary Material.

Figure 2 depicts anecdotal evidence on the behaviour of the component-wise expectation of the loss gradient as more samples from the posterior distribution are incorporated into the BNN predictive distribution. Similarly to how in Figure 1 for the half-moons dataset we observe that the gradient of the loss goes to zero when increasing number of training points and number of parameters, here we have that, as the number of samples taken from the posterior distribution of $\mathbf{w}$ increases, all the components of the gradient approach zero. Notice that the gradient of the individual NNs (that is those with just one sampled weight), is far away from being zero. As shown in Theorem 1, it is only through the Bayesian averaging of ensemble predictor that the gradients cancel out.

This is confirmed in Figure 3, where we provide a systematic analysis of the aggregated gradient convergence properties on 1k test images for MNIST and Fashion-MNIST. Each dot shown in the plots represents a component of the expected loss gradient from each one of the images, for a total of 784k points used to visually approximate the empirical distribution of the component-wise expected loss gradient. For both HMC and VI the magnitude of the gradient components drops as the number

of samples increases, and tends to stabilize around zero already with 100 samples drawn from the posterior distribution, suggesting that the conditions laid down in Theorem 1 are approximately met by the BNN analysed here. Notice that the gradients computed on HMC trained networks drops more quickly toward zero. This is in accordance to what is discussed in Section 4, as VI introduces additional approximations in the Bayesian posterior computation.

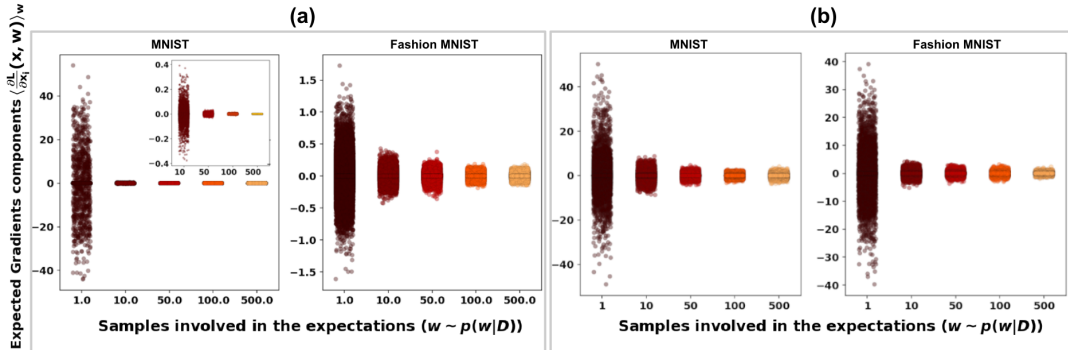

Figure 3: The components of the expected loss gradients approach zero as the number of samples from the posterior distribution increases. For each number of samples, the figure shows 784 gradient components for 1k different test images, from both the MNIST and Fashion MNIST datasets. The gradients are computed on HMC (a) and VI (b) trained BNNs.

## 5.2 Gradient-Based Attacks for BNNs

The fact that gradient cancellation occurs in the limit does not directly imply that BNN predictions are robust to gradient-based attacks in the finite case. For example, FGSM attacks are crafted such that the direction of the manipulation is given only by the sign of the expectation of the loss gradient and not by its magnitude. Thus, even if the entries of the expectation drop to an infinitesimal magnitude but maintains a meaningful sign, then FGSM could potentially produce effective attacks. In order to test the implications of vanishing gradients on the robustness of the posterior predictive distribution against gradient-based attacks, we compare the behaviour of FGSM and PGD[4] attacks to a randomly devised attack. Namely, the random attack mimics a randomised version of FGSM in which the sign of the attack is sampled at random. In practice, we perturb each component of a test image by a random value in $\{-\epsilon, \epsilon\}$.

In Table 1 we compare the effectiveness of FGSM, PGD and of the random attack. We report the adversarial accuracy (i.e. probability that the network returns the ground truth label for the perturbed input) for 500 images. For each image, we compute the expected gradient using 250 posterior samples. The attacks were performed with $\epsilon = 0.1$. In almost all cases, we see that the random attack outperforms the gradient-based attacks, showing how the vanishing behaviour of the gradient makes FGSM and PGD attacks uneffective. For all attacks we used the categorical cross-entropy loss function

| Dataset/Method | Rand | FGSM | PGD |
|---|---|---|---|
| MNIST/HMC | **0.850** | 0.960 | 0.970 |
| MNIST/VI | 0.956 | **0.936** | 0.938 |
| Fashion/HMC | **0.812** | 0.848 | 0.826 |
| Fashion/VI | **0.744** | 0.834 | 0.916 |

Table 1: Adversarial robustness of BNNs trained with HMC and VI with respect to the *random attack* (Rand), FGSM and PGD.

which is related to the likelihood used during training. Furthermore, in Table 2 in the Supplementary we also run the same evaluation for when the same network employed in Table 1 is trained with SGD and deep ensembles. In both cases both FGSM and PGD are effective, suggesting how simply model averaging and mini-batches are not enough to achieve a robust model.

## 5.3 Robustness Accuracy Analysis in Deterministic and Bayesian Neural Networks

In Section 4, we noticed that as a consequence of Theorem 1, high accuracy might be related to high robustness to gradient-based attacks for BNNs. Notice, that this would run counter to what has been observed for deterministic NNs trained with SGD [Su et al., 2018]. In this section, we look at an array of more than 1000 BNNs with different hyperparameters trained with HMC and VI on MNIST and Fashion-MNIST.[5] We experimentally evaluate their accuracy/robustness trade-off on FGSM attacks as compared to that obtained with deterministic NNs trained via SGD based methods. For the robustness evaluation we consider the average difference in the softmax prediction between the original test points and the crafted adversarial input, as this provides a quantitative and smooth measure of adversarial robustness that is closely related with mis-classification ratios [Cardelli et al., 2019a]. That is, for a collection of $N$ test point, we compute $\frac{1}{N} \sum_{j=1}^{N} |\langle f(\mathbf{x}_j, \mathbf{w}) \rangle_{p(\mathbf{w}|D)} - \langle f(\tilde{\mathbf{x}}_j, \mathbf{w}) \rangle_{p(\mathbf{w}|D)}|_{\infty}$.

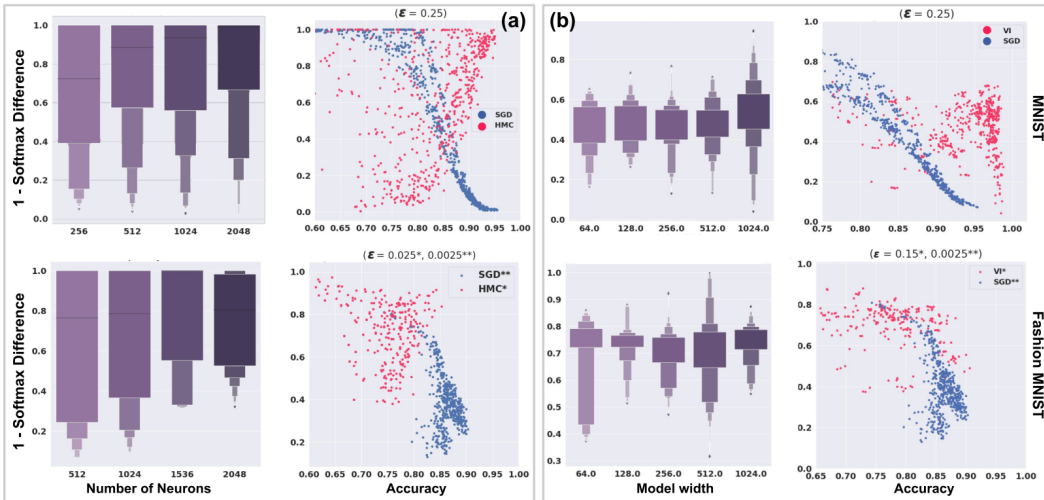

Figure 4: Robustness-Accuracy trade-off on MNIST (first row) and Fashion MNIST (second row) for BNNs trained with HMC (a), VI (b) and SGD (blue dots). While a trade-off between accuracy and robustness occur for deterministic NNs, experiments on HMC show a positive correlation between accuracy and robustness. The boxplots show the correlation between model capacity and robustness. Different attack strength ($\epsilon$) are used for the three methods accordingly to their average robustness.

The results of the analysis are plotted in Figure 4 for MNIST and Fashion MNIST. Each dot in the scatter plots represents the results obtained for each specific network architecture trained with SGD (blue dots), HMC (pink dots in plots (a)) and VI (pink dots in plots (b)). As already reported in the literature [Su et al., 2018] we observe a marked trade-off between accuracy and robustness (i.e., 1 - softmax difference) for high-performing deterministic networks. Interestingly, this trend is fully reversed for BNNs trained with HMC (plots (a) in Figure 4) where we find that as networks become more accurate, they additionally become more robust to FGSM attacks as well. We further examine this trend in the boxplots that represent the effect that the network width has on the robustness of the resulting posterior. We find the existence of an increasing trend in robustness as the number of neurons in the network is increased. This is in line with our theoretical findings, i.e., as the BNN approaches the over-parametrised limit, the conditions for Theorem 1 are approximately met and the network is protected against gradient-based attacks. On the other hand, the trade-off behaviours are less obvious for the BNNs trained with VI and on Fashion-MNIST. In particular, in plot (b) of Figure 4 we find that, similarly to the deterministic case, also for BNNs trained with VI, robustness seems to have a negative correlation with accuracy. Furthermore, for VI we observe that there is some trend dealing with the size of the model, but we only observe this in the case of VI trained on MNIST where it can be seen that model robustness may increase as the width of the layers increases, but this can also lead to poor robustness (which may be indicative of mode collapse).

# 6    Conclusions

The quest for robust, data-driven models is an essential component towards the construction of AI-based technologies. In this respect, we believe that the fact that Bayesian ensembles of NNs can evade a broad class of adversarial attacks will be of great relevance. While promising, this result comes with some significant limitations. First and foremost, performing Bayesian inference in large non-linear models is extremely challenging. While in our experiments cheaper approximations such as VI also enjoyed a degree of adversarial robustness, albeit reduced, there are no guarantees that this will hold in general. To this end, we hope that this result will spark renewed interest in the pursuit of efficient Bayesian inference algorithms. Secondly, our theoretical results hold in a thermodynamic limit which is never realised in practice. More worryingly, we currently have no rigorous diagnostics to understand how near we are to the limit case, and can only reason about this empirically. We notice here that several other studies [Bekasov and Murray, 2018, Li and Gal, 2017, Feinman et al., 2017, Rawat et al., 2017] have focused on pointwise uncertainty to detect adversarial behaviour; while this does not appear relevant in the limit scenario, it might be a promising indicator of robustness in finite data conditions. Thirdly, we have focused on two attack strategies which directly utilise gradients in our empirical evaluation. More complex gradient-based attacks, such as [Carlini and Wagner, 2016, Papernot et al., 2017, Moosavi-Dezfooli et al., 2016], as well as non-gradient based/ query-based attacks, also exist [Ilyas et al., 2018, Wicker et al., 2018]. Evaluating the robustness of BNNs against these attacks would also be interesting.

Finally, the proof of our main result highlighted a profound connection between adversarial vulnerability and the geometry of data manifolds; it was this connection that enabled us to show that principled randomisation might be an effective way to provide robustness in the high dimensional context. We hope that this connection will inspire novel algorithmic strategies which can offer adversarial protection at a cheaper computational cost.

# 7    Broader Impact

This work is a theoretical investigation in the large data limit of vulnerability of Bayesian Neural Networks to gradient-based attacks. The main result is that, in this limit, BNNs are not vulnerable to such attacks, as the input gradient vanishes in expectation. This advancement provides a theoretically-provable rational for selecting BNNs in applications where there is concern about attackers performing fast, gradient-based attacks. However, it does not provide any guarantee on the actual safety of BNNs trained on a finite amount of data. Our work may positively benefit the study of adversarial robustness for BNNs and the investigation of properties that make these networks less vulnerable than deterministic ones. These features could then potentially be transferred to other network paradigms and lead to greater robustness of machine learning algorithms in general. However, there may still exist different attacks leading BNNs to misclassifications and our contribution does not provide any defence technique against them.

In the last few years adversarial examples have presented a major hurdle to the adoption of AI systems in any security related field, whose applications go from self-driving vehicles to medical diagnoses. Machine learning algorithms show remarkable performance and generalization capabilities, but they also manifest weaknesses that are not consistent with human understanding of the world. Ultimately, the lack of knowledge about the difference between human and machine interpretation of reality leads to an issue of public trust. The development of procedures that are robust to changes in the output and that represent calibrated uncertainty, would inherently be more trust-worthy and allow for wide-spread adoption of deep learning in safety and security critical tasks.

# 8    Funding Disclosure

This project was partially funded by the EU's Horizon 2020 program under the Marie Sklodowska-Curie grant agreement No 722022 "AffecTec" and by the Italian PRIN project "SEDUCE" No 2017TWRCNB.

## Footnotes

[1]The code for the experiments can be found at: `https://github.com/ginevracoal/robustBNNs`.

[2]For simplicity we omit the dependence of the loss from the ground truth.

[3]Both a uniform distribution and a wide Gaussian distribution act as uninformative priors.

[4]with 15 iterations and 1 restart.

[5]Details on the NN architectures used can be found in the Supplementary Material.

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
