[Supplementary Material]

# Additional theory background and proofs

## 9 Global convergence of over-parameterised DNNs

We briefly recapitulate here the main results on global convergence of over-parameterised neural networks [Du et al., 2018, Mei et al., 2018, Rotskoff and Vanden-Eijnden, 2018]. We follow more closely the notation of Rotskoff and Vanden-Eijnden [2018] and reference to that paper for more formal proofs and definitions.

The setup of the problem is as follows: we are using a NN $f(\mathbf{x}, \mathbf{w})$ to approximate a function $\tilde{f}(\mathbf{x})$. The target function is observed at points drawn from a data distribution $p(D)$ while the weights of the NN are drawn from a measure $\mu(\mathbf{w})$. The support of the data distribution $p(D)$ is the *data manifold* $\mathcal{M}_D \subset \mathbb{R}^d$. The discrepancy between the observed target function and the approximating function is measured through a suitable loss function $L(\mathbf{x}, \mathbf{w})$ which needs to be a convex function of the difference between observed and predicted values (e.g. squared loss for regression or cross-entropy loss for classification).

These results require a set of technical but rather standard assumptions on the target function and the NN units (assumptions 2.1-2.4 and 3.1 in Rotskoff and Vanden-Eijnden [2018]), which we recall here for convenience:

- The input and feature space are closed Riemannian manifolds, and the NN units are differentiable.
- The unit is *discriminating*, i.e. if it integrates to zero when multiplied by a function $g$ for all values of the weights, then $g = 0$ a.e. .
- The network is sufficiently expressive to be able to represent the target function.
- The distribution of the data input is not degenerate (Assumption 3.1).

One can then prove the following results:

- The loss function is a convex functional of the measure on the space of weights.
- Training a NN (with a finite number of units/ weights) by gradient descent approximates a gradient flow in the space of measures. Therefore, by the Law of Large Numbers, gradient descent on the exact loss (infinite data limit) converges to the global minimum (constant zero loss) when the number of hidden units grows to infinity (overparameterised limit).
- Stochastic gradient descent also converges to the global minimum under the assumption that every minibatch consists of novel examples.

In other words, Rotskoff and Vanden-Eijnden [2018] show that the celebrated Universal Approximation Theorem of Cybenko [1989] is realised dynamically by stochastic gradient descent in the infinite data/ overparameterised limit.

## 10 Proofs of technical results

We provide here additional details for the theoretical results in the main text. We will assume that the assumptions of Rotskoff and Vanden-Eijnden [2018] hold, as recalled in Section 9.

**Lemma 3.** *Let $f(\mathbf{x}, \mathbf{w})$ be a fully trained overparametrized NN on a prediction problem with a.e. smooth data manifold $\mathcal{M}_D \subset \mathbb{R}^d$. Let $\mathbf{x}^* \in \mathcal{M}_D$ s.t. $B_d(\mathbf{x}^*, \epsilon) \subset \mathcal{M}_D$, with $B_d(\mathbf{x}^*, \epsilon)$ the $d$-dimensional ball centred at $\mathbf{x}^*$ of radius $\epsilon$ for some $\epsilon > 0$. Then $f(\mathbf{x}, \mathbf{w})$ is robust to gradient-based attacks at $\mathbf{x}^*$ of strength $\leq \epsilon$ (i.e. restricted in $B_d(\mathbf{x}^*, \epsilon)$).*

**Proof.** By the results of Rotskoff and Vanden-Eijnden [2018], Mei et al. [2018], Du et al. [2018] we know that, in the large data limit, an overparametrized NN will achieve zero loss on the data manifold once fully trained. By assumption, the data manifold contains a whole open ball centred at $\mathbf{x}^*$, so the loss will be constant (and zero) in an open neighbourhood of $\mathbf{x}^*$. Consequently, the loss gradient at $\mathbf{x}^*$ will be zero in a whole open neighbourhood of $\mathbf{x}^*$; therefore, any attack based on moving the input point in the direction of the gradient at $\mathbf{x}^*$ or a nearby point (such as PGD) will fail to change the input and consequently fail to change the output value, thus guaranteeing robustness. $\square$

**Corollary 2.** *Let $f(\mathbf{x}, \mathbf{w})$ be a fully trained overparametrized NN on a prediction problem with data manifold $\mathcal{M}_D \subset \mathbb{R}^d$ smooth a.e. (where the measure is given by the data distribution $p(D)$). If $f$ is vulnerable to gradient-based attacks at $\mathbf{x}^* \in \mathcal{M}_D$ in the infinite data limit, then a.s. $\dim(\mathcal{M}_D) < d$ in a neighbourhood of $\mathbf{x}^*$.*

**Proof.** If $f$ is vulnerable at $\mathbf{x}^*$ to gradient based attacks, then the gradient of the loss at $\mathbf{x}^*$ must be non-zero. By Lemma 1 we know that, if the data manifold $\mathcal{M}_D$ has locally dimension $d$, then the gradient has to be zero. Hence $\dim(\mathcal{M}_D) < d$ in a neighbourhood of $\mathbf{x}^*$. $\qquad\square$

**Lemma 4.** *Let $f(\mathbf{x}, \mathbf{w})$ be a fully trained overparametrized NN on a prediction problem on data manifold $\mathcal{M}_D \subset \mathbb{R}^d$ a.e. smooth. Let $\hat{\mathbf{x}} \in \mathcal{M}_D$ to be attacked and let the normal gradient at $\hat{\mathbf{x}}$ be $\mathbf{v_w}(\hat{\mathbf{x}}) := \nabla_{\perp \mathbf{x}} L(\hat{\mathbf{x}}, \mathbf{w})$ be different from zero. Then, in the infinite data limit and for almost all $\hat{\mathbf{x}}$, there exists a set of weights $\mathbf{w}'$ such that*

$$f(\mathbf{x}, \mathbf{w}') = f(\mathbf{x}, \mathbf{w}) \text{ a.e. in } \mathcal{M}_D,$$

$$\nabla_{\perp \mathbf{x}} L(\hat{\mathbf{x}}, \mathbf{w}') = -\mathbf{v_w}(\hat{\mathbf{x}}).$$

**Proof.** By assumption, the function $f(\mathbf{x}, \mathbf{w})$ is realisable by the NN and therefore differentiable. To show that there exists (at least) one set of weights that lead to a function satisfying the constraints in (4) and (5), we proceed by steps. First, we observe that the loss is a functional over functions $g \colon \mathcal{M}_D \to [0, 1]$, given explicitly by

$$L[g] = \int_{\mathcal{M}_D} dq \sum_y p(y|\theta) \log g(\theta)$$

where $\theta$ is a parametrisation on $\mathcal{M}_D$, and we have written the data generating distribution $p(D) = p(y|\theta)q(\theta)$ as the product of the distribution of input values times the class conditional distribution. However, evaluating the loss over a function $\phi : \mathbb{R}^d \to [0, 1]$ defined over the whole ambient space only makes sense if one also defines a projection from the ambient space into the data manifold. It is however still possible, given a function defined over the whole ambient space, to define the loss computed on its restriction over $\mathcal{M}_D$ and the normal gradient to the manifold by using the ambient space metric and the decomposition it induces of the tangent space into directions along $\mathcal{M}_D$ and directions orthogonal. Normal derivatives of $L[\phi(\mathbf{x})]$ can then be defined as standard. For any function $\phi(\mathbf{x})$ on $\mathcal{M}_D$ the normal gradient of the loss function[6] is

$$\nabla_{\perp \mathbf{x}} L(\phi(\mathbf{x})) = \frac{\delta L(\phi)}{\delta \phi} \nabla_{\perp \mathbf{x}} \phi(\mathbf{x})$$

Assuming the functional derivative of the loss is a differentiable function, as is the case e.g. with cross-entropy, then condition 5 can be rewritten as

$$h(\phi(\hat{\mathbf{x}}), \nabla_{\perp \mathbf{x}} \phi(\hat{\mathbf{x}})) = 0 \tag{6}$$

for a suitably smooth function $h$.

To construct a function $\phi$ that satisfies both conditions (4) and (5), we assume that the data manifold admits smooth local coordinates in an open ball $\mathcal{M}_D \cap B_d(\hat{\mathbf{x}}, \epsilon)$ of radius $\epsilon$ centred at $\hat{\mathbf{x}}$ (which is true for almost all points by assumption). We then define $\phi(\mathbf{x}) = f(\mathbf{x}, \mathbf{w}) + g(\mathbf{x})$, where $g(\mathbf{x})$ is smooth, supported in $B_d(\hat{\mathbf{x}}, \epsilon)$ and zero on the boundary of the ball $\partial B_d(\hat{\mathbf{x}}, \epsilon)$, and $g(\mathbf{x}) = 0 \quad \forall \mathbf{x} \in \mathcal{M}_D \cap B_d(\hat{\mathbf{x}}, \epsilon)$. Therefore, $\phi$ satisfies condition (4) by construction. In particular we can impose condition (4) on $g$ in the local coordinates around $\hat{\mathbf{x}}$, by using a slice chart on $\mathcal{M}_D \cap B_d(\hat{\mathbf{x}}, \epsilon)$.

In the overparametrized limit, it will always be possible to approximate the resulting function $\phi$ by choosing suitable weights $\mathbf{w}'$ for the NN, thus proving the Lemma. Notice that condition 6 holds on a fixed point $\hat{\mathbf{x}}$ under attack, hence at different attack points we may in principle have different $w'$ satisfying the lemma.

$\qquad\square$

# 11 Comparison with Deep Ensembles

Deep ensembles, as proposed by Lakshminarayanan et al. [2017], are an ensemble of neural networks trained from different randomly selected initial conditions, which are then averaged in order to make a prediction. In Table 2 we consider the same network used to perform the experiments in Section 5.2 (hyper-parameters are reported in Table 4) and run a comparison with both deterministic NNs and deep ensembles. As expected, Bayesian NNs are more robust than deterministic ones. Moreover, we find that deep ensembles and deterministic NNs are comparable in terms of robustness, suggesting that simply averaging predictions for different weight initialization and mini-batching is not enough to achieve a robust model.

| Model | Test accuracy | FGSM accuracy | PGD accuracy |
|---|---|---|---|
| Deterministic NN | 97.69 | 21.19 | 1.45 |
| Ensemble NN | 99.4 | 20.6 | 0.3 |
| Bayesian NN | 96.1 | 90.0 | 89.8 |

Table 2: FGSM and PGD attacks on the network employed in Section 5.2. We compare a deterministic NN, a deep ensemble NN (of size 100), and a BNN (trained with VI). Attacks are performed on 1k test points from the MNIST dataset. We observe that VI trained network achieve better robustness against PGD and FGSM.

# 12 Training hyperparameters for BNNs

**Half moons grid search**

| Posterior samples | {250} |
|---|---|
| HMC warmup samples | {100, 200, 500} |
| Training inputs | {5000, 10000, 15000} |
| Hidden size | {32, 128, 256, 512} |
| Nonlinear activation | Leaky ReLU |
| Architecture | 2 fully connected layers |

Figure 5: Hyperparameters for training BNNs in Figure 1

**Training hyperparameters for HMC**

| Dataset | MNIST | Fashion MNIST |
|---|---|---|
| Training inputs | 60k | 60k |
| Hidden size | 1024 | 1024 |
| Nonlinear activation | ReLU | ReLU |
| Architecture | Fully Connected | Fully Connected |
| Posterior Samples | 500 | 500 |
| Numerical Integrator Stepsize | 0.002 | 0.001 |
| Number of steps for Numerical Integrator | 10 | 10 |

Table 3: Hyperparameters for training BNNs using HMC in Figures 2 and 3.

**Training hyperparameters for VI**

| Dataset | MNIST | Fashion MNIST |
|---|---|---|
| Training inputs | 60k | 60k |
| Hidden size | 512 | 1024 |
| Nonlinear activation | Leaky ReLU | Leaky ReLU |
| Architecture | Convolutional | Convolutional |
| Training epochs | 5 | 10 |
| Learning rate | 0.01 | 0.001 |

Table 4: Hyperparameters for training BNNs using VI in Figures 2 and 3.

**HMC MNIST/Fashion MNIST grid search**

| | |
|---|---|
| Posterior samples | {250, 500, 750*} |
| Numerical Integrator Stepsize | {0.01, 0.005*, 0.001, 0.0001} |
| Numerical Integrator Steps | {10*, 15, 20} |
| Hidden size | {128, 256, 512*} |
| Nonlinear activation | {relu*, tanh, sigmoid} |
| Architecture | {1*,2,3} fully connected layers |

Table 5: Hyperparameters for training BNNs with HMC in Figure 4. * indicates the parameters used in Table 1 of the main text.

**SGD MNIST/Fashion MNIST grid search**

| | |
|---|---|
| Learning Rate | {0.001*} |
| Minibatch Size | {128, 256*, 512, 1024} |
| Hidden size | {64, 128, 256, 512, 1024*} |
| Nonlinear activation | {relu*, tanh, sigmoid} |
| Architecture | {1*,2,3} fully connected layers |
| Training epochs | {3,5*,7,9,12,15} epochs |

Table 6: Hyperparameters for training BNNs with SGD in Figure 4. * indicates the parameters used in Table 1 of the main text.

**SGD MNIST/Fashion MNIST grid search**

| | |
|---|---|
| Learning Rate | {0.001, 0.005, 0.01, 0.05} |
| Hidden size | {64, 128, 256, 512} |
| Nonlinear activation | {relu, tanh, sigmoid} |
| Architecture | {2, 3, 4, 5} fully connected layers |
| Training epochs | {5, 10, 15, 20, 25} epochs |

Table 7: Hyperparameters for training BNNs with SGD in Figure 4.

## Footnotes

[6]Notice that this is only defined on the data manifold $\mathcal{M}_D$, while $\mathbf{x}$ is a coordinate system in the ambient space $\mathbb{R}^d$.