[Reviews · NeurIPS 2020]

Review 1

Summary and Contributions: I've read the rebuttal and discussions. My opinion hasn't changed substantially and I will leave my overall score unchanged. - The basic idea is that the posterior average of the loss function has a zero $x$ (input) gradient in the limit of a large amount of data. - The paper claims that this makes the Bayesian setting largely immune to gradient based adversarial attacks.

Strengths: - I found the result very interesting. - The paper does a reasonable effort to experimentally justify the result.

Weaknesses: - A frustrating aspect is that the paper is not particularly self contained and requires the reader to understand the previous specialised works in this area. - For example, definitions such as "overparamaterised" or "fully trained" are not explained directly in the text. - It is also not particularly intuitive (to me at least) from the text why the result is true. However, I believe there is simpler and more intuitive explanation available.

Correctness: - This seems fine.

Clarity: - Whilst the paper is well written, I found it hard to understand to what class of problems the results apply. It would have been I think much easier if the author(s) had explained some of the conditions they refer to in previous works. - It's also unclear to me why a much simpler and intuitive (and perhaps more general) explanation isn't possible for the central result -- see the additional feedback.

Relation to Prior Work: - This seems fine.

Reproducibility: Yes

Additional Feedback: It's not clear to me why a much simpler derivation isn't possible. An informal argument could be simply: - The Bernstein-von Mises theorem says that the posterior will tend to a Gaussian with mean $\mu$ at the maximum likelihood optimum and covariance $\Sigma$ given by the scaled inverse Fisher matrix. - Consider a loss function $L(y,prediction)$, with prediction given by $f(x,w)$ where $x$ is the input and $w$ the parameters of the neural network. The expected loss is then (using reparameterisation) $ E[L(y,f(x,w))]_{p(w|data)} \rightarrow E[L(y,f(x,\mu+C\epsilon)]_{\epsilon\tilde N(0,I)}$ where $C$ is the Cholesky matrix of $\Sigma$. Since $C$ is small, one can Taylor expand (essentially in $C\epsilon) so that the leading contribution is (since the linear term in \epsilon has zero mean) $L(y,f(x,\mu))$ Taking the gradient wrt $x$ gives $\partial_a L(y,a) \partial_x f(x,\mu)$, evaluated at $a=f(x,\mu)$ For a realisable problem in which the data is generated by the model, $y=f(x,w_*) where $w_*$ is the true parameter that generated the data. Hence, the leading term is $L'(y=f(x,w_*),a=f(x,\mu))$ where $L'$ is the gradient wrt the second argument of $L$. The second term is zero (due to averaging) and the third term (corresponding to the second derivative in the Taylor expansion) will scale $O(1/n$) where $N$ is the number of training datapoints. Provided the ML estimator converges to the true parameter $w_*$ then this term will be zero, since for any meaningful loss function the loss will be minimal when the prediction and observed value match. A special case of the above that is easy to compute fully is when $f$ is a linear function of $w$ since all integrals are exact. However, the above holds more generally provided the Bernstein-von Misses and ML conditions are met. What this means is that this would hold for any "sensible" prior for which the Berstein-von Mises result holds and any sensible function. I'm not clear why one cannot use such an elementary (and more general?) approach. Are there technical subtleties that would prevent this?


Review 2

Summary and Contributions: This paper studies the robustness of Bayesian neural networks (BNNs) to (gradient-based) adversarial attacks both theoretically and empirically. Firstly, the authors prove that in the limit of large data and overparametrization, gradient-based adversarial attacks are effective due to the data lying on a lower-dimensional manifold. They then prove that in that limit, BNN posteriors are robust to such attacks. Lastly, these theoretical findings in the limiting case are supported by an empirical study on the MNIST and Fashion MNIST datasets, demonstrating that even in practical settings, BNNs (trained with both Hamiltonian Monte Carlo and Variational Inference) are robust to gradient-based adversarial attacks, while also maintaining high accuracy.

Strengths: This work tackles an important open problem, i.e. studying the adversarial robustness of Bayesian neural networks, and presents novel and insightful results of both theoretical and empirical nature. To my knowledge, this is the first paper that derived theoretical adversarial robustness guarantees for BNNs, which I think is an important result for future papers to build upon, despite its limitations described below. Both the theoretical analysis and empirical evaluation appear to be sound. The limitations and shortcomings of the work are discussed in detail; I highly appreciate this honesty. The authors further describe the consequences of their theoretical results, conjecturing to which degree their results might still hold if certain assumptions are not met in practice, which is insightful. This work is of high relevance to the NeurIPS community, and makes significant and likely impactful contributions that will be interesting to a fairly large audience in machine learning. Overall, I think that these strenghts outweigh the weaknesses described below, which is why I am in favor of having this paper accepted.

Weaknesses: The paper does a fair and thorough job of describing some of its main limitations (see Section 6), which includes that 1) doing Bayesian inference in large-scale models (which is required for their results to hold) remains challenging in practice (which I personally don't really consider to be a shortcoming of their paper), 2) their theoretical results only hold for a limiting case that cannot be attained in practice, and 3) they only consider certain types of adversarial attacks, namely those relying on gradient information. Another shortcoming is that the empirical evaluation is somewhat small-scale, only considering shallow fully-connected neural networks and simple benchmark datasets (i.e. MNIST and Fashion MNIST). While the experimental results are fairly conclusive and convincing in those settings, it is not clear if those results extend to larger-scale, practically relevant scenarios, i.e. settings in which we care most about adversarial robustness. It would significantly strenghten the paper if it included empirical results on networks that are deeper, have more complex structure (e.g. convolutional layers), and that are evaluated on more challenging benchmark tasks (e.g. CIFAR10/0, or even ImageNet). While I understand that Bayesian inference remains challenging for such networks, recent advances in Bayesian deep learning, e.g. (Ritter 2018, Maddox 2019, Osawa 2019, Zhang 2020), have succeeded (to some degree) in scaling different types of inference to larger-scale settings. Also, the experiments only consider image classification tasks; while I understand that those are the most commonly-used benchmarks for general-purpose machine learning methods, it would nevertheless be insightful to see results for other application domains in which adversarial robustness is important. Furthermore, in the experiments described in Section 5.2, it would be insightful to compare to an ordinary (i.e. non-Bayesian / point-estimated / deterministic) neural network as a baseline. In particular, it would be good to know how the robustness of a BNN compares to that of a non-Bayesian NN on the considered adversarial attacks. At present, this experiment does not support the conclusion that BNNs are more robust to gradient-based adversarial attacks than deterministic NNs (which is not a claim that the authors seem to make explicitly, although it would be very intersting to know whether this is the case). Questions: 1) in Section 5, you first mention that "it is not obvious (and likely not true) that the posterior distribution is the sole ensemble with the zero averaging property of the gradients", before saying that "while the Bayesian posterior ensemble may not be the only randomization to provide protection, it is clear that some simpler randomizations such as bootstrap will be ineffective"; how do deep ensembles (Lakshminarayanan, 2017) fit into this picture, which also provide an approximation to the Bayesian model average in Equation (1) (Wilson and Izmailov, 2020), and which can be fit using the bootstrap (Lakshminarayanan, 2017), but are in practice more commonly fit by solely relying on the randomness within the weight initialization and mini-batching? given the effectiveness and thus popularity of deep ensembles for Bayesian model averaging, it would be insightful to have a brief discussion on how your results might or might not translate to them; furthermore, I think deep ensembles would be an interesting additional baseline in your empirical evaluation, which you should consider adding 2) in Section 5, you mention that "unless the priors are too informative, we do not expect a major deviation from the idealised case"; do you think of the commonly-used standard-normal prior N(0,1) as an informative or uninformative prior? which conditions exactly does the prior need to fulfil in order for your result to hold? what is the main challenge in extending your result to other types of priors? 3) do you have any intuition for why in deterministic NNs, accuracy and robustness appears to be negatively correlated, why in BNNs, it seems to be positively correlated (as you demonstrate in Section 5.3, and as previous work has also partially shown)? References: Lakshminarayanan et al., "Simple and scalable predictive uncertainty estimation using deep ensembles", NeurIPS 2017. Ritter et al., "A scalable laplace approximation for neural networks", ICLR 2018. Maddox et al., "A simple baseline for bayesian uncertainty in deep learning", NeurIPS 2019. Osawa et al., "Practical deep learning with bayesian principles", NeurIPS 2019. Zhang et al., "Cyclical stochastic gradient MCMC for Bayesian deep learning", ICLR 2020. Wilson and Izmailov, "Bayesian deep learning and a probabilistic perspective of generalization", arXiv 2020.

Correctness: Yes, the theoretical claims and empirical methodology appears to be correct, as far as I can tell.

Clarity: Yes, the paper is overall very well written and thus easy to follow. The material is described in a way such that readers which are neither familiar with Bayesian neural networks nor adversarial attacks should be able to understand the main results. The theoretical claims are presented clearly, and the main ideas behind the proofs are laid out in an intuitive way. There are only minor issues impacting clarity, which are listed under "Additional feedback" below.

Relation to Prior Work: Yes, the paper describes all relevant papers (that I am aware of) studying the robustness of BNNs to adversarial examples, and contrasts them to this work.

Reproducibility: No

Additional Feedback: POST-REBUTTAL: Thank you for your response! After reading all other reviews and the rebuttal, my sentiment remains, and I am still in favour of seeing the paper accepted. ============== Here is a list of minor issues, mainly affecting clarity and presentation, which I hope will be helpful to the authors when revising their manuscript: - according to the NeurIPS formatting instructions, "all headings should be lower case (except for first word and proper nouns)" (https://media.neurips.cc/Conferences/NeurIPS2020/Styles/neurips_2020.pdf, Section 3); this does not hold for all your headings - l. 38: "BNNs architectures" --> "BNN architectures" - l. 61: It is not clear why the sentence starts with "However"; it would be good to describe why exactly it is a problem that these approaches used MC dropout - l. 62: "as a posterior inference" --> "as a posterior inference method" or "for/to do posterior inference" - Section 2: as this section covers two distinct and independent topics (i.e., Bayesian Neural Networks and Adversarial Attacks, respectively), it would aid readability and clarity to explicitly structure it as such by introducing two subsections, i.e. "2.1 Bayesian Neural Networks" and "2.2 Adversarial Attacks" - l. 71: the citation to (Neal, 2012) seems misplaced; perhaps better to place it at the end of the sentence, or, if you want to keep the position, _before_ the comma in "In the NN scenario," - l. 72: "then" can be removed - Eq. (1): the notation of the index i is inconsistent, i.e., non-boldface underneath the summation operator, and boldface when indexing the weight vector w - l. 85: "is" --> "defines/yields"; or, perhaps clearer, "Equation (1) defines the \emph{predictive distribution} of the BNN induced by the ensemble of NNs." - l. 92: "in x" --> "at x" - l. 92-93: add commas --> "In the context of BNNs, where attacks are against the predictive distribution of Eqn (1), FGSM becomes" - l. 95: "Expression for" --> "The expressions for" - l. 95: "or" --> "and" - l. 97: "directly look at" --> "specifically look at/focus on" - l. 101: here you refer to the equation as "Equation (x)", while in l. 93, you referred to an equation as "Eqn (x)", which is inconsistent - l. 106: "which proved that" --> "which proved" - l. 108: "in supplementary material" --> "in the supplementary material"; also, it would be helpful to refer to a specific section in the supplementary material - l. 111: "outside it" --> "outside of it" - l. 111: again, it would be helpful to refer to a specific section in the supplementary material - l. 120: "of fully trained BNN" --> "of a fully trained BNN" - l. 125: "with B() the d-dimensional ball" --> "with B() being the d-dimensional ball" or "where B() is the d-dimensional ball" - l. 134-135: here x^* is non-boldface, which is inconsistent with before - l. 154: there's unnecessary whitespace between "uniformative" and the superscript 2; also, it's perhaps better to refer to the footnote after "prior", rather than after "uninformative" - Footnote 2: "gaussian" --> "Gaussian"; also, how wide does the Gaussian have to be to count as uninformative? - l. 156: "It remains therefore to be proved the" --> "Therefore, what remains to be proved is the" - l. 159: by "Let \hat{x} to be attacked", do you mean that \hat{x} is the starting point of the attack to be perturbed, or that \hat{x} itself is the attack? perhaps this could be clarified by slightly reformulating it - l. 159-160: "let the normal gradient at \hat{x} be v[...] be different from zero" --> "let the normal gradient at \hat{x}, v[...], be different from zero" - l. 178: "the network's parameters" --> "the network's number of parameters" - Figure 1: "loss gradients components" --> "loss gradient components" - Figure 1: the Figure is not entirely clear; it should be made clearer that the number on top of each image (i.e. 32, 128, 256, 512) denotes the number of hidden neurons, e.g. by using titles "X hidden neurons"; why are there multiple dots for a fixed number of neurons and a fixed number of training inputs? what do you mean by "Each dot represents a different NN architecture"? from the supplementary material, it seems that you are using the same 2-layer leaky ReLU architecture, with only the number of hidden neurons differing? also, in the axis labels, do you abuse notation, where <.>_w is equivalent to <.>_{p(w|D)}? why not write it out? - l. 195: it would be useful to have at least one citation for the claim that it is an "often repeated mantra that bootstrap is an approximation to Bayesian inference" - l. 196: "don't" --> "do not" - l. 207: "can be find" --> "can be found" - l. 207: "Supplementary Material" --> supplementary material; also, it would be helpful reference a specific section in the supplementary material - l. 210: "settings" --> "setting" - l. 213: "achieved" --> "achieve" - l. 215: "Supplementary Material" --> supplementary material; again, it would be helpful reference a specific section - l. 217-218: "prediction distribution" --> "predictive distribution" - l. 219: "zero increasing number" --> "zero when increasing the number" - l. 222: "null" --> "zero" - Figure 2: the fontsizes could be larger in the Figure to aid readability - l. 225: here you write "Fashion-MNIST", while previously you wrote "Fashion MNIST" - l. 226: "represent" --> "represents" - l. 229: "tend" --> "tends" - l. 231: "here analysed" --> "analysed here" - l. 231: "drops" --> "drop" - l. 233: "on" --> "in" or "to" - Figure 3: in the x-axes in Figure (a), you might want to use integers to denote the number of samples, as you did in Figure (b) - Figure 3: what is the small subfigure within the MNIST part of Figure (a)? - l. 239: "maintaining" --> "maintain" - Footnote 3: this should preferrably be a full sentence - l. 257: what do you mean by "related to the likelihood"? is it not equivalent to the likelihood? what's the difference? - l. 260: "Notice, that this would run counter to what" --> "Notice that this would contradict what" - l. 262: how is it more than 1000 different BNN architectures? looking at the supplementary material, you only seem to vary between 3-5 values for the hidden size, 3 different activation functions, and 3 values for the number of layers (the others are not really architectural parameters); if your count also includes the other parameters, perhaps writing "an array of more than 1000 BNNs with different hyperparameters" would be more accurate - l. 263: again, you write "Fashion-MNIST" as opposed to "Fashion MNIST" (pick one throughout for consistency) - Footnote 4: "Supplementary Material" --> supplementary material; again, it would be helpful reference a specific section - l. 268: "test point" --> "test points" - l. 279: here you spell it "over-parameterised", while previously you were spelling it "overparametrized"; choose one for consistency - l. 280: "gradients attack" --> "gradient-based attacks" - l. 286: use either "also" or "as well", but not both - l. 286: not so clear how this relates to mode collapse; a slightly longer explanation would help - Figure 4: "occur" --> "occurs"; "strength" --> "strengths"; "accordingly" --> "according"; also, in the boxplots in Figure (b), the x-axis label should again be integers, as in Figure (a); moreover, in Figure (a), you call it "Number of Neurons", while in Figure (b), you call it "model width"; aren't these the same quantities? furthermore, it would be useful to summarize the conclusion from the boxplots in the caption; finally, it's not so clear how exactly those boxplots were computed, and what they visually represent (I haven't seen this form of plot before); perhaps it would be useful to explain those plots in a bit more detail - l. 292: not sure what "in our hands" is supposed to mean; perhaps reformulate to use a more common phrase? - l. 304-305: "would also be interesting" --> "would be an interesting avenue for future work"


Review 3

Summary and Contributions: This paper presents a theoretical study of the effectivity of Bayesian neural networks to defend gradient-based adversarial attacks in the large-data, overparametrized limit. The accuracy and robustness of the Bayesian posteriors to such attacks are demonstrated by several experiments.

Strengths: The strengths of the paper are 1. the interesting justification for the finding of the effectivity of BNN in the cancelation of the gradient of the loss function with respect to the data, in expectation with respect to the posterior of the parameters. 2. the convincing demonstration by several experiments for BNN trained by both Hamiltonian Monte Carlo and variational inference, for several datasets including MNIST, Fashion MNIST, and Half Moon.

Weaknesses: The weaknesses are: 1. The main contribution of the theoretical proof does not seem to be original/novel, which is essentially built on the work mentioned in the paper. Moreover, the proof of Theorem 1 does not seem to be complete to me in that the normal gradient in Lemma 2 is not shown to be zero in expectation with respect to the posterior. 2. It is not clear if the samples obtained by HMC and VI are distributed according to the posterior, for such high-dimensional inference problems. Can the author provide some justification?

Correctness: The claims and empirical study seem to be correct.

Clarity: The paper is well written and easy to follow.

Relation to Prior Work: The discussion of how this work is related to prior work is clear.

Reproducibility: No

Additional Feedback: It is not clear on the setup and property (learning rate, termination criteria, independence, etc.) of HMC and VI samples.


Review 4

Summary and Contributions: This paper analyze the geometry of adversarial attacks arises as a result of degeneracy in the data distribution, and demonstrate in the limit BNN posteriors are robust to gradient-based adversarial attacks. With experiments on MNIST and Fashion MNIST using HMC and VI, they show the robustness and accuracy of BNNs.

Strengths: 1. The authors proved the expectation of the gradient is 0 on a fully trained overparameterized BNN with reasonable assumption. 2. The authors linked their proof with several recent findings with BNN. 3. They authors ran empirical experiments to show 1) the zero-averaging gradient, 2) The robustness of BNNs w.r.t. different type of attacks, 3) the robustness-accuracy relationship.

Weaknesses: 1. In the proof, the assumption is still relative too strong, far from reality. 2. Some of the explanation of recent work on BNN by applying their framework are somewhat ad hoc. 3. In the experiments, there is lacking of insights why HMC has better performance than VI in MNIST but on the contrary in Fashion MNIST. Also it lacks some comparison of training BNN using methods other than HMC and VI.

Correctness: It is correct.

Clarity: It is well written, though I will argue the plots have minimized fonts which impair the reading experience with a normal scale.

Relation to Prior Work: Some of the work relating to train BNN are missing.

Reproducibility: No

Additional Feedback:

[Author Response · NeurIPS 2020]

**R1) Comments on the main proof strategy.** We thank the reviewer for the insightful comments on the proof. We agree that the sketched argument based on the Bernstein-von Mises theorem is simpler, yet it relies - at least in its vanilla version- on the assumption that the target function is representable in a fixed parametric model. In order to avoid this assumption - which is not controllable a priori, we rely on limit results in the over-parameterised regime, in which the size of the model is larger than the input data. In this scenario, both the input data and the model size are taken to the limit, and so the limiting model is non-parameteric and guaranteed to have zero loss on the data manifold. It is not obvious to us if and under which conditions the BvM theorem generalises to this setting, but this is indeed a promising direction to investigate further. We still believe, though, that our proof provides some interesting geometric insights on adversarial attacks. We will clarify better in the main text notions like "overparamaterise" or "fully trained".

**R2, R3, R4) Results reproducibility, convolutional layers, and number of different architectures.** We remark that the hyper-parameters used for training are reported in the Supplementary (Section 3). The source code will be made available after the review phase. We remark that our experiments comprises both fully connected (up to 5 layers) and convolutional layers (see Table 2). Noted by R2, Table 4 in the supplementary was accidentally truncated by one value in several rows, we will update the parameters accordingly (in total we trained 1728 HMC BNNs).

**R2) Using deterministic networks and deep ensembles as baseline models.** We agree with the reviewer and in Table 1 we consider the same NN used to perform the experiments in Section 5.2 (hyper-parameters are reported in Table 3 in the Supplementary) and run a comparison with both deterministic NNs and deep ensembles (Lakshminarayanan, 2017). We further evaluate the robustness of deep ensembles on a subset of the NNs employed in Section 5.3. We find that deep ensemble NNs have a robustness similar to that of deterministic NNs suggesting that simply averaging predictions for different weight initialization and mini-batching is not enough to achieve a robust model. We will add these results in the main text.

**R2) Priors.** In our proof setting, an uninformative prior is one that gives equal density to all the possible weights realisations. This can be seen as the limit of a Gaussian with infinite variance. In practice, the relative importance of the prior w.r.t. the likelihood diminishes as more data are used for training, and the posterior distribution gets pulled apart from the prior. In the experiments reported in the paper, we have found an $\mathcal{N}(0,1)$ prior to work well, as the posterior variance gets to around $0.05$ after training. We performed evaluations with a higher prior variance (up to $10$) and noticed a similar behaviour of the loss gradients.

| Model | Test accuracy | FGSM accuracy | PGD accuracy |
|---|---|---|---|
| Deterministic NN | 97.69 | 21.19 | 1.45 |
| Ensemble NN | 99.4 | 20.6 | 0.3 |
| Bayesian NN | 96.1 | 90.0 | 89.8 |

Table 1: FGSM and PGD attacks on the network employed in Section 5.2. We compare a deterministic NN, a deep ensamble NN (of size $100$), and a BNN (trained with VI). Attacks are performed on 1k test points from the MNIST dataset. We observe that VI trained network achieve better robustness against PGD and FGSM.

**R2) Correlation between accuracy and robustness.** When a BNNs has a high number of neurons and high accuracy the conditions for Theorem 1 are approximately met. This guarantees that the network is protected against gradients attack, thanks to the cancelling effect of Theorem 1. For deterministic NNs Theorem 1 does not hold. The trade-off between robustness and accuracy in that case has been already observed and studied (Zhang et al.2019).

**R3) Proof of Th 1: novelty and completeness.** We would like to stress that Theorem 1 and its proof are novel. In fact, although we rely on known results for over-parametrised NNs, to the best of our knowledge, the application of these results in the context of robustness of Bayesian NNs and Lemma 2 are novel. Furthermore, we would like to clarify that for an input point and a fully trained model, Lemma 2 guarantees that there exists another model such that the NN has the same loss on the data manifold and opposite orthogonal gradients on that input point. Hence, by definition, the NN will have same likelihood on models. If they also have same prior (uninformative prior assumption) then they also have same posterior. This entails the cancellation of orthogonal gradient average.

**R3) Are HMC and VI distributed according to the posterior?** Unfortunately, for non-linear networks computation of the posterior is analytically intractable. Nevertheless, HMC converges to the true posterior in the limit of infinitely many Monte Carlo samples taken (we used 500 samples in our experiments). On the other side, VI is a more scalable approximate inference method, but has no convergence guarantees to the true posterior. This also explains why for MNIST, where both HMC and VI obtain $> 90\%$ accuracy, HMC tends to be more robust than VI.

**R4) Why HMC performs better than VI in MNIST but not in FashionMNIST?** HMC converges to the true posterior, but it is less scalable than VI. As a result, on MNIST where HMC is able to achieve $> 90\%$ accuracy, it tends to be more robust than VI. On the other hand, on FashionMNIST we were not able to train a BNN with HMC to have high accuracy, hence, we are far from the regime required by our Theorem 1 to achieve cancelling gradients.

**R4) Add comparison with other training methods.** Please, see R2) Using deterministic network and deep ensembles as baseline models.

[Meta-Review · NeurIPS 2020]

The proposed approach using Bayesian neural networks for robustness to adversarial is timely and interesting. One of the reviewers proposed an outline for a much simpler proof than what is used in the paper. Other reviewers raised concerns about the fairness of the comparison between HMC and VI, arguing that VI is not necessarily more scalable. Please account for reviewer comments (including updates after the response) in performing revisions.